# Super-resolution imaging reveals α-synuclein seeded aggregation in SH-SY5Y cells

Jason C. Sang [1,2], Eric Hidari [1,2], Georg Meisl [1], Rohan T. Ranasinghe[1], Maria Grazia Spillantini[3] & David Klenerman [1,2✉]

Aggregation of α-synuclein (α-syn) is closely linked to Parkinson's disease (PD) and the related synucleinopathies. Aggregates spread through the brain during the progression of PD, but the mechanism by which this occurs is still not known. One possibility is a self-propagating, templated-seeding mechanism, but this cannot be established without quantitative information about the efficiencies and rates of the key steps in the cellular process. To address this issue, we imaged the uptake and seeding of unlabeled exogenous α-syn fibrils by SH-SY5Y cells and the resulting secreted aggregates, using super-resolution microscopy. Externally-applied fibrils very inefficiently induced self-assembly of endogenous α-syn in a process accelerated by the proteasome. Seeding resulted in the increased secretion of nanoscopic aggregates (mean 35 nm diameter), of both α-syn and Aβ. Our results suggest that cells respond to seed-induced disruption of protein homeostasis predominantly by secreting nanoscopic aggregates; this mechanism may therefore be an important protective response by cells to protein aggregation.

[1] Department of Chemistry, University of Cambridge, Cambridge, UK. [2] UK Dementia Research Institute at Cambridge, Cambridge, UK. [3] Department of Clinical Neuroscience, University of Cambridge, Cambridge, UK. ✉email: dk10012@cam.ac.uk

Parkinson's disease (PD) and the related synucleinopathies, such as dementia with Lewy bodies and multiple system atrophy, are progressive neurodegenerative disorders. The primary hallmark of synucleinopathies is the deposition of intracellular inclusions in the brain, known as Lewy bodies. These deposits consist of aggregated α-synuclein (α-syn)[1], a 140-amino-acid protein that is ubiquitously expressed in neurons[2]. The spread of α-syn aggregates through brain tissues drives the progression of these diseases; it is currently unclear how this spreading happens.

Mounting evidence implies that α-syn, like other aggregation-prone proteins such as tau and prion protein, exerts pathogenicity by a prion-like process[3–5]. In this scenario, α-syn aggregates act as templates, or "seeds", by recruiting α-syn monomers into fibrils; these then fragment into two or more aggregates, which in turn seed further aggregation. Both fibril growth and fragmentation are therefore needed for seed replication. The disease spreads when aggregates travel from cell to cell, and seed aggregation of endogenous α-syn in the recipient cells. Small soluble aggregates of α-syn, rather than large inclusions, are believed to be the key species in α-syn neurotoxicity and spread between cells[6,7]. Recombinant α-syn aggregates generated by sonicating fibrils, or preformed fibrils (PFFs) can induce endogenous α-syn aggregation and spread through non-transgenic mouse brain after injection[8], while soluble α-syn aggregates seed aggregation in cell lines and primary neuronal cultures[9], as well as in mice[10].

Although we have previously shown that aggregation of recombinant α-syn follows a prion-like, fragmentation-driven pathway in vitro[11,12], the rates of aggregation and fragmentation will differ in vivo. For example, the cellular clearance machinery may affect the kinetics of aggregation by modulating the concentrations of key species in the reaction pathway. The ubiquitin-proteasome pathway (UPP) is normally linked to the clearance of small aggregates as well as monomeric α-syn turnover[13], whereas large aggregates are mainly processed by the autophagy-lysosomal pathway (ALP)[14]. In PD and related synucleopathies, the progressive accumulation of α-syn aggregates impairs both UPP and ALP[15–17]; the disruption of either pathway can also lead to α-syn accumulation[18]. Aggregated Aβ, tau, and prion protein inhibit the proteasome in mice[19–21], while we recently showed that the proteasome can fragment aggregated proteins into smaller toxic aggregates[22]. In the cell, the proteasome may therefore amplify the number of α-syn seeds in the early stages of aggregation by fragmenting fibrils.

It is not known how α-syn aggregates spread between cells. One suggestion is that α-syn aggregates are secreted to the extracellular space, then transferred to neighboring cells[23–25]. Soluble α-syn aggregates have been found in the extracellular environment[26,27], and they are efficiently secreted via exosomes and then taken up by recipient cells[28]. Moreover, an α-syn antibody can reduce neuronal and glial accumulation of α-syn and cell-to-cell transmission[29]. These findings together suggest that extracellular α-syn aggregates, especially small species, seed aggregation of intracellular α-syn and potentially play an important role in the spread of α-syn pathology in vivo. This might result from templated seeding after uptake, but aggregates can also replicate indirectly, by inducing an inflammatory response which leads to cellular aggregation[12,30].

The potential interplay of so many cellular pathways makes it crucial to follow replication of aggregates in cells to understand whether prion-like spreading plays an important role in PD. However, protein aggregates in cells are structurally heterogeneous, typically smaller than the diffraction limit of visible light (~250 nm), and drastically outnumbered by monomers, which limits the insight gained from conventional fluorescence microscopy; it has, to date, prevented quantitative experiments to be performed. We previously developed aptamer DNA-PAINT (AD-PAINT)[31,32] to specifically super-resolve aggregated α-syn and Aβ in cell lines and iPSC-derived neurons with a high spatial resolution of ~25 nm. Here, we use a conformational antibody (MJFR-14-6-4-2) that preferentially recognizes aggregated α-syn[33], to super-resolve cytoplasmic α-syn aggregates in undifferentiated SH-SY5Y neuroblastoma cells using conventional antibody-DNA-PAINT (Figs. S1–S3 and Supplementary Movies 1–4). Imaging the evolution of cytoplasmic and extracellular α-syn aggregates after the uptake of preformed seeds allowed us to investigate how the proteasome and autophagy modulate seed replication, and how neuroinflammation affects α-syn aggregation. Our quantitative data shed light on the efficiency of these different steps in replication of α-syn aggregates, suggesting that inflammation plays a dominant role in spreading aggregates between cells.

## Results

**Exogenous α-syn fibrils, but not monomers, can trigger endogenous α-syn aggregation.** Firstly, we established conditions for cellular uptake of dye-labeled α-syn seeds (mean = 35 ± 2 nm in length, Fig. S4), generated by sonicating PFFs. Conventional epifluorescence microscopy showed that these sonicated fragments were taken up by SH-SY5Y cells after 4 h incubation, which was enhanced by the lipid-based mediator, Bioporter (Fig. S5). We then used this protocol to study whether externally added, unlabeled seeds triggered aggregation of endogenous α-syn in cells with no protein overexpression, using super-resolution (SR) microscopy (Fig. 1a). After 4 h of incubation with seeds, cells were extensively rinsed with PBS to remove residual adsorbed seeds (defined as time = 0 h, or T0) and then incubated further. At T0, we typically detected 0–10 α-syn aggregates per cell (Fig. 1b, c). These small numbers of seeds taken up by cells suggest that the subsequent accumulation of α-syn aggregates results from aggregation of endogenous α-syn. After 24 h (T24), we detected intracellular α-syn aggregates after transduction with preformed fibrillar seeds (Fig. 1b), but not monomeric α-syn or PBS (Fig. S6), confirming that fibrillar seeds can indeed template and trigger aggregation of endogenous α-syn in SH-SY5Y cells. We also note that without Bioporter, seeding (as detected at T24) is extremely inefficient (Fig. S7): the seeding probability is estimated to be $10^{-8}$, so that about $10^8$ encounters with PFFs are necessary to seed a cell (see "Methods" for estimation). To test for seed-induced cytotoxicity, we tracked cell numbers and the release of lactate dehydrogenase (LDH) after transduction (Fig. S8). The results showed no substantial cytotoxicity, and only a small difference in the growth rates of unseeded and seeded cells. Our protocol therefore seeds cells without apparent cell death.

To gain more insights, we followed the growth of intracellular α-syn aggregates for 72 h after seeding (Fig. 1b), acquiring around 30 SR images of individual cells at defined time points. After seeding, an increasing number of cells developed aggregates over the measurement period: the percentage of cells with <10 aggregates dropped from ~80% to 10% within 72 h, while the proportion of cells with >50 aggregates increased from 0 to 70% (Fig. 1f). Together with increases in aggregate length and aggregates per cell (Figs. 1c, d and S9), these data reveal rapid generation and growth of α-syn aggregates after transduction. We note that the number of aggregates varied between cells and that a fraction of cells did not develop aggregates during incubation. This can be explained by cells being at different stages of the cell cycle, or the fact that the increase in cell numbers over time (Fig. S8b), meaning that some cells at each time point have recently undergone cell division. A third possibility is that only a fraction of the cells was initially seeded and then seeds spread to neighboring cells over time.

We then extracted rate constants for seeded α-syn aggregation in cells by fitting the data in Fig. 1c to a kinetic model (Fig. 1e; see

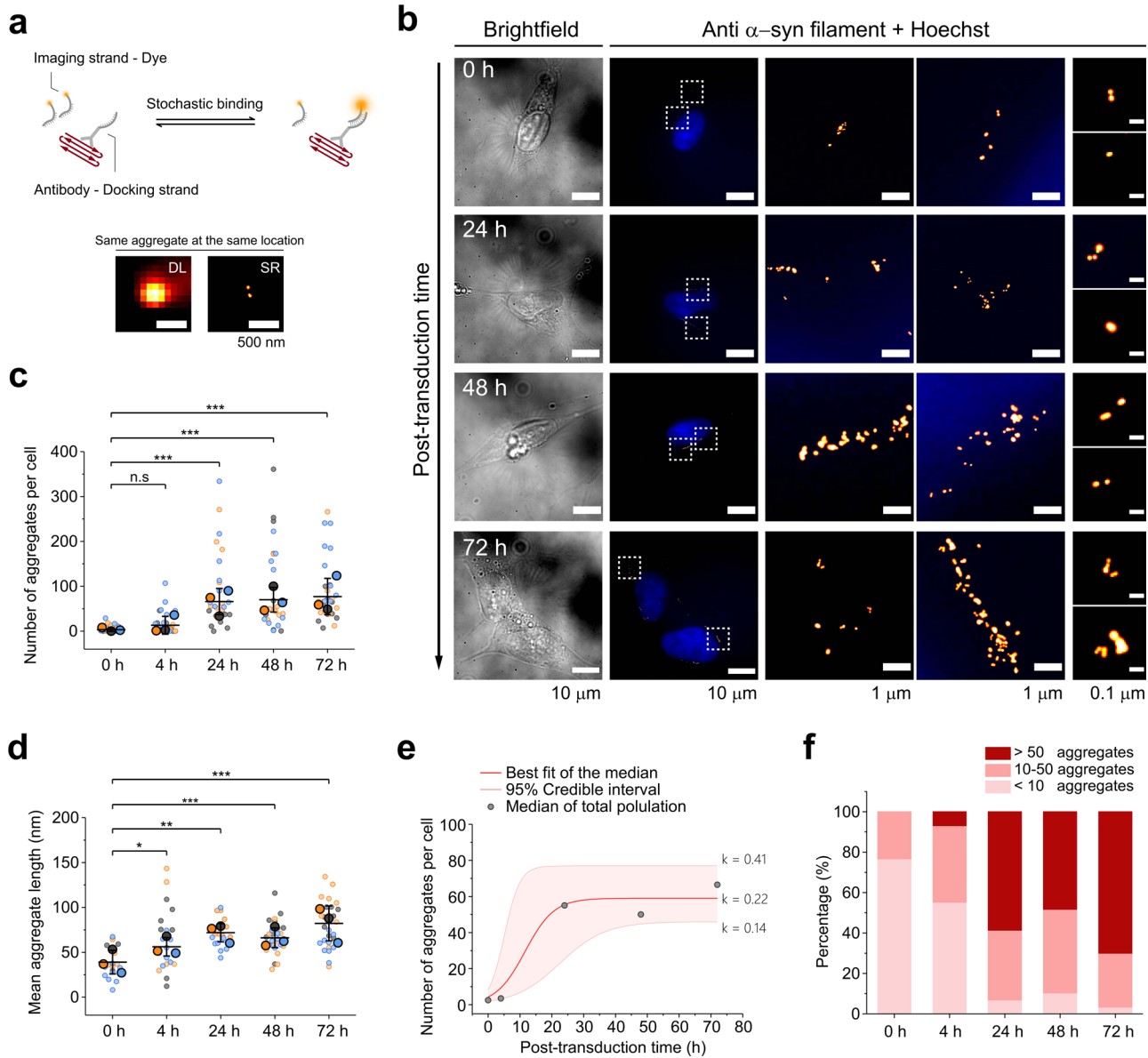

**Fig. 1 Super-resolving the replication of intracellular α-syn aggregates inside cells. a** Schematic of antibody-DNA-PAINT imaging. Cells at defined time points were immunostained with MJFR-14-6-4-2 conjugated with the docking strand (DS). The complementary imaging strand (IS), linked to a fluorescent Cy3b dye, interacts with the docking strand stochastically at a nanomolar concentration and hence can "paint" the boundary of the given object. The two microscopic images show that super-resolved (SR) images can distinguish two aggregated particles compared with conventional diffraction-limited (DL, thioflavin T-stained) images. **b** Growth of intracellular α-syn aggregates over time. SH-SY5Y cells are incubated with a mixture of PFF seeds (final 2.5 μM monomer equivalent of sonicated α-syn aggregates) and Bioporter for 4 h, then rinsed with PBS and cultured with fresh DMEM/10% FBS (time = 0 h after rinsing). Nuclei are stained blue with Hoechst dye; super-resolved α-syn aggregates are shown in red. **c, d** Intracellular aggregates increase in number and length after seed transduction. Statistical difference was determined by one-way ANOVA with Tukey's multiple comparison test. N (total number of cells imaged) = 30, 29, 29, 29, 30 for 0, 4, 24, 48, 72 h, respectively. The cells were separately pooled for each biological replicate (n = 3) and the median calculated for each pool; those three medians were then used to calculate the mean (horizontal bars), standard deviation (error bars), and p values. *p < 0.05, **p < 0.01, ***p < 0.001, n.s. not significant; the difference of the means is not significant at the 0.05 level. **e** Kinetic analysis of the formation of intracellular aggregates; the median values from **c** are fitted to a minimal model of replication (see "Methods" for details). The best fit for the replication rate within cells is 0.22 h$^{-1}$, with 95% credible interval CI between 0.14 and 0.41. **f** The percentage of cells containing >50 aggregates grows over time, while the percentage of cells containing <10 aggregates falls.

"Methods" for details). The time required for the number of aggregates to double was calculated and used to extract rate constants for fibril growth and fragmentation. The doubling time for α-syn aggregates in SH-SY5Y cells was ~5 h (95% credible interval = 2.4–7.1 h), with a replication rate of 0.22 h$^{-1}$ (95% CI = 0.14–0.41 h$^{-1}$). We used the doubling time to estimate the rates for fibril growth and fragmentation rates in SH-SY5Y

cells of 0.01–10 s$^{-1}$ and $7 \times 10^{-10}$–$7 \times 10^{-7}$ s$^{-1}$ (see "Methods" for details).

**Proteasomal activity accelerates the formation of endogenous α-syn aggregates, while clearance by autophagy is inefficient.** We then asked how protein clearance pathways contribute to the

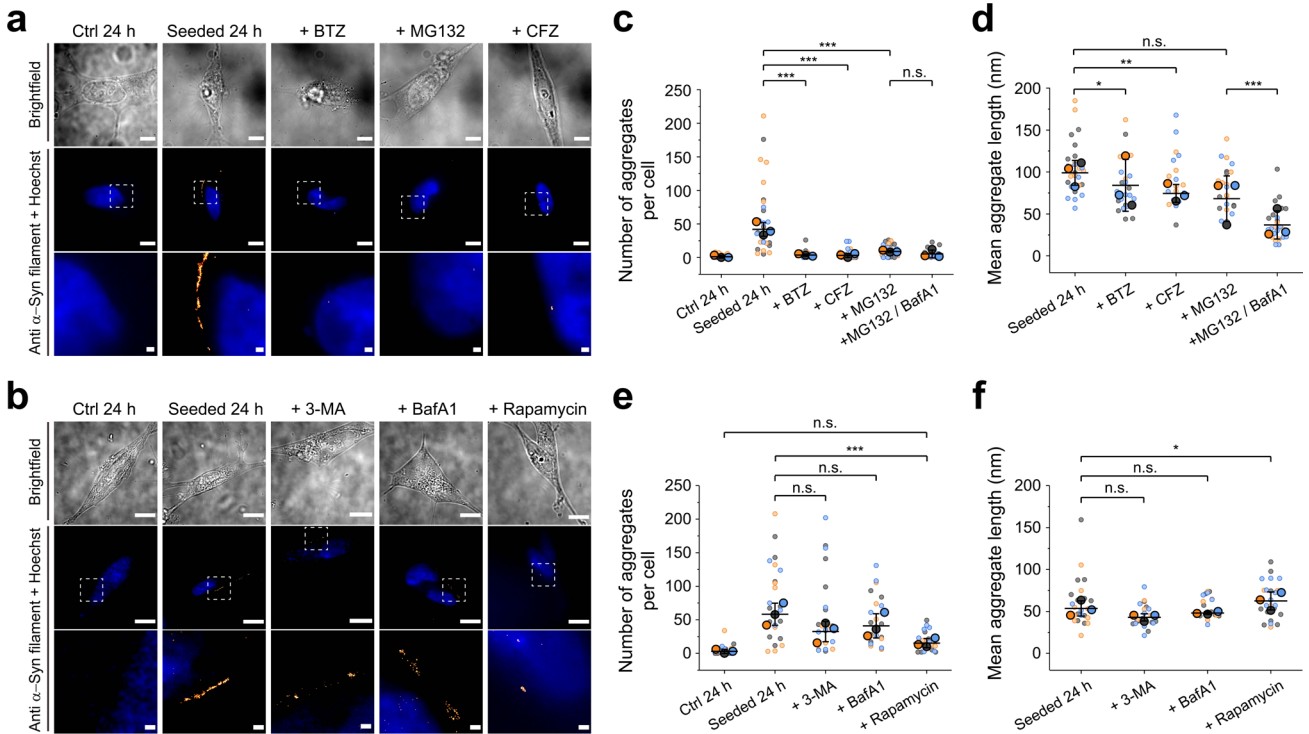

**Fig. 2 The protein clearance machinery shapes α-syn aggregation in SH-SY5Y cells.** After seeding with PFFs, SH-SY5Y cells were incubated DMEM/ 10% FBS supplemented with proteasomal and/or lysosomal modulators for 24 h, then α-syn aggregates detected with antibody-DNA-PAINT. BTZ: Bortezomib (5 μM); CFZ: Carfilzomib (1 μM); MG132 (1 μM); 3-MA: 3-Methyladenine (2.5 mM); BafA1: Bafilomycin A1 (0.4 μM); rapamycin (60 nM). **a** Proteasomal inhibition reduces α-syn aggregation in SH-SY5Y cells. **b** Autophagy-lysosomal pathway does not remove α-syn efficiently in SH-SY5Y cells. Nuclei are stained blue with Hoechst dye; super-resolved α-syn aggregates are shown in red. Inhibiting the proteasome decreased the number of aggregates per cell (**c**) but not the aggregate length (**d**). In contrast, inhibiting autophagy did not affect the number of α-syn aggregates (**e**) but stimulating autophagy with rapamycin restored protein clearance; neither treatment affected aggregate length (**f**). Statistical difference was determined by one-way ANOVA with Tukey's multiple comparison test. The cells were separately pooled for each biological replicate ($n = 3$) and the median calculated for each pool; those three medians were then used to calculate the mean (horizontal bars), standard deviation (error bars), and $p$ values. *$p < 0.05$, **$p < 0.01$, ***$p < 0.001$, n.s. not significant; the difference of the means is not significant at the 0.05 level. For proteasomal inhibition, $N = 30, 29, 29, 30, 28, 27$ for Ctrl 24 h, Seeded 24 h, +BTZ, +MG132, +CFZ, +MG132/BafA1, respectively, from three biological replicates. For autophagy inhibition/induction, the total number of cells imaged $N = 21, 24, 24, 22, 23$ for Ctrl 24 h, Seeded 24 h, +3-MA, +BafA1, +Rapamycin, respectively, from three biological replicates. Scale bars in **a** and **b** (from top to bottom): 10, 10, and 1 μm.

early stages of seeded α-syn aggregation in cells, starting with the proteasomes. To answer this question, we separately treated cells with three proteasome inhibitors, bortezomib (BTZ), carfilzomib (CFZ), and MG132, after seed transduction. Remarkably, the cellular level of α-syn aggregates after 24 h incubation with inhibitors was reduced by 5–15-fold (Figs. 2a, c, d and S10). Kinetic analysis of these data estimates a slowing of the doubling time for α-syn aggregation by at least tenfold when the proteasome is inhibited.

These results support the hypothesis that proteasomal activity could accelerate α-syn accumulation in cells, consistent with our recent observation that the proteasome dissembles large α-syn and tau fibrils into smaller, more cytotoxic species in vitro[22]; these species might template further aggregation. However, proteasome inhibition can activate the ALP in cells[34,35], so the reduction in α-syn accumulation we saw may result from increased ALP activity. To test this, we inhibited both ALP and UPP using Bafilomycin A1 (BafA1) combined with MG132. BafA1 is a macrolide antibiotic that inhibits the vacuolar H+ ATPases, thereby blocking acidification and lysosomal degradation in general[36]. After 24 h of BafA1/MG132 treatment, the level of α-syn did not increase compared with that of MG132-treated cells (Fig. 2c). This result indicates the reduction of the α-syn aggregation is due to proteasome inhibition rather than increased ALP activity or the interplay between ALP and UPP.

In order to better understand the effects of UPP and ALP in seeded α-syn aggregation, we investigated the role of autophagy (Figs. 2b, e, f and S11), using two inhibitors, 3-methyladenine (3-MA) and BafA1, and one enhancer, rapamycin, added after seed transduction. It is important to note that although 3-MA is widely used to block autophagosome formation by inhibiting class III PI3K, prolonged exposure (like the 24 h used here) in a nutrient-rich medium can alternatively promote autophagy by inhibiting class I PI3K, which is an upstream regulator of mTOR complex[37]. On the other hand, rapamycin activates the ALP by inhibition of mTOR. Neither 3-MA nor BafA1 treatment altered the level of cytoplasmic α-syn after seed transduction, while inducing autophagy with rapamycin reduced the α-syn level. These results suggest that the ALP could not efficiently remove cytoplasmic α-syn aggregates, due to being either impaired or overwhelmed by the numbers of α-syn aggregates, unless activated by rapamycin. This is also consistent with the conclusion from our dual-inhibition experiments (Fig. 2c) that inefficiency of ALP does not contribute to the decreased α-syn aggregation under proteasomal inhibition.

**Small α-syn aggregates are secreted to the extracellular environment.** The increasing proportion of cells containing aggregates after exposure to PFFs (Fig. 1f) led us to investigate whether α-syn

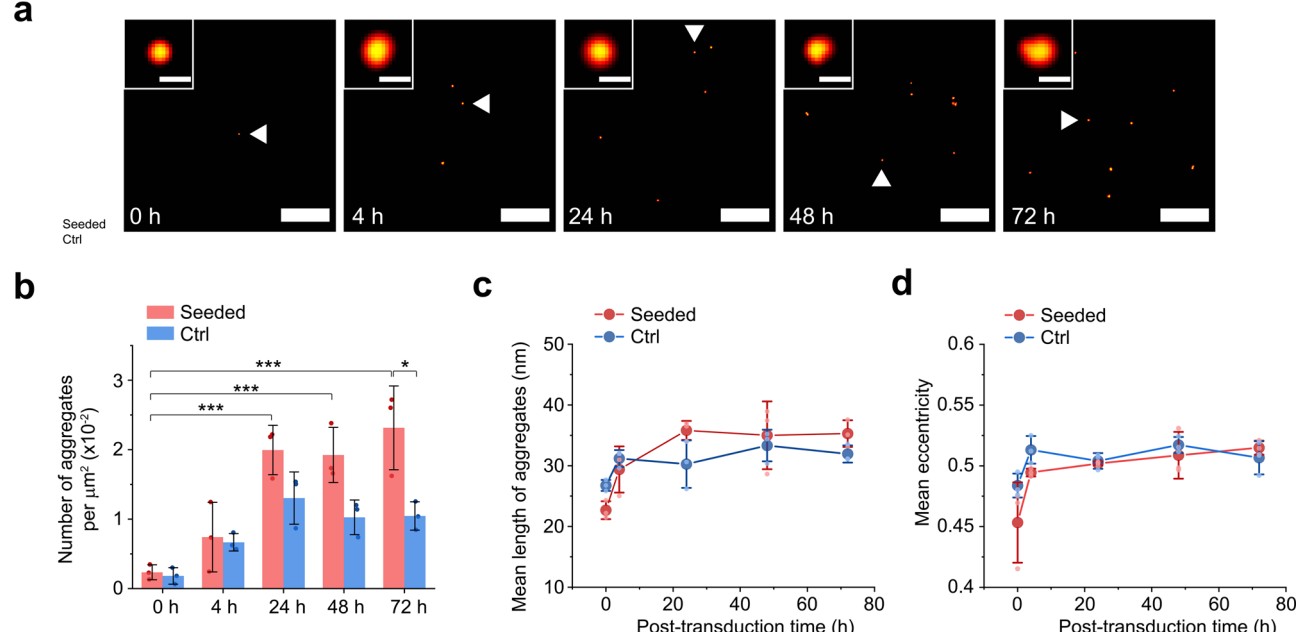

**Fig. 3 Seeded cells secrete small aggregates.** Culture medium was collected at defined time points after seeding and imaged with AD-PAINT. **a** Super-resolution images of extracellular aggregates. Secreted aggregates accumulate over time following seeding (**b**), while their length (**c**) and eccentricity (**d**) increase slightly. Statistical difference was determined by two-way ANOVA with Tukey's multiple comparison test. *$p < 0.05$, **$p < 0.01$, ***$p < 0.001$. Mean ± SD from three biological replicates. Scale bars in **a**: 500 nm (main panels) and 25 nm (insets).

seeds spread in a prion-like manner. This process may occur via cellular secretion of seeds and subsequent uptake by recipient cells[25,38], but it is unclear whether enough aggregates travel between cells in this way to initiate seeding. We characterized secreted α-syn aggregates with AD-PAINT, a form of DNA-PAINT that uses a conformation-specific DNA aptamer. It was not possible to image aggregates in cell medium using antibody-DNA-PAINT, due to excess non-specific adsorption onto glass coverslips that resulted in false-positive signals. Despite its lower specificity—AD-PAINT recognizes aggregates of both Aβ and α-syn—measurements inside SH-SY5Y cells gave similar results to antibody-DNA-PAINT (Fig. S12). Seeded cells and unseeded cells both released increasing numbers of aggregates over time (Fig. 3a, b). The apparent mean length of aggregates increased modestly over the course of the measurement (from 23 ± 1 to 35 ± 1 nm), a size range close to the precision of AD-PAINT (Figs. 3c and S13). The eccentricity of aggregates, on the other hand, did not change significantly; therefore the aggregates retained similar morphology.

Since the DNA aptamer recognizes aggregates of both Aβ and α-syn, we next performed immunodepletion using Aβ and α-syn antibodies to gauge the proportions of the both proteins in the culture medium. IgG antibodies 6E10 and Syn211 were used to deplete Aβ or α-syn aggregates in the culture medium separately (Fig. 4a), while a control IgG1 antibody was used to examine potential non-specific antibody recognition in the set-up and did not change the mean number of aggregates from medium only controls. Consistent with Figs. 3c and S11, we detected very few aggregates larger than 75 nm in length (Fig. 4b), and aggregate length did not change substantially after immunodepletion (Fig. S14). The immunodepletion of aggregates compared to control medium predominantly results from removal of Aβ (α-syn: 16%, Aβ: 84% at T72 control medium), while in the seeded medium, Aβ and α-syn are more evenly depleted (α-syn: 33%, Aβ: 67% at T72 seeded medium). It is interesting to note that, comparing with the control medium, the amount of Aβ aggregates is higher by 2.5-fold in the seeded medium, while the amount of α-syn aggregates shows an 8.5-fold increase.

Based on the PFF calibration curve (Fig. S15), the aptamer-detected aggregate concentration was 270 nM at T72, while at T0 it was below the detection range. The secreted aggregates were much smaller than cytoplasmic aggregates (35 ± 1 nm vs. 82 ± 11 nm at T72). The estimated concentration was consistent with our ELISA results with the seeded medium (397 ± 248 nM at T72 and 0.27 ± 0.05 nM at T0; monomer equivalent; Fig. S16).

Our data imply that the $10^6$ cells in each culture dish on average secrete ~$10^6$ α-syn aggregates per cell per day to the extracellular space, assuming that an aggregate consists of 30 monomers. This assumption is supported by in vitro data, where spherical α-syn oligomers are reported to be 10–30 nm[39], consisting of 30 monomers on average[40]. In contrast, the number of intracellular α-syn aggregates on average is 75 aggregates per cell as shown in Fig. 1c. Therefore, the secreted α-syn aggregates outnumber the intracellular aggregates by ~$10^5$ fold, suggesting that cells secreting small aggregates is an important protective mechanism, which also occurs in unseeded cells at a low level.

## Discussion

Our single-aggregate experiments provide quantitative data on seeding by α-syn in neuroblastoma cells. We have summarized the major findings as a schematic in Fig. 5. This work yielded several surprising insights, showing that the cellular environment alters the rates of templated seeding from those measured in the test tube[11,12]. We observe a fast aggregation of α-syn in cells after seeding, and the process slows down after T24. The number of intracellular aggregates depends on the difference between the rate of aggregate production and removal. Aggregate growth by addition of monomers and subsequent fragmentation increases the number of aggregates, while aggregate degradation by the cellular machinery and secretion to the extracellular environment decreases the number of aggregates. The plateau of seeded aggregation can be explained by the cells reaching a steady state after an initial fast increase due to seeding. The rate of seed production is roughly balanced by the rate of seed removal due to up-regulation of the cellular machinery, such as ALP, aggregate secretion into the

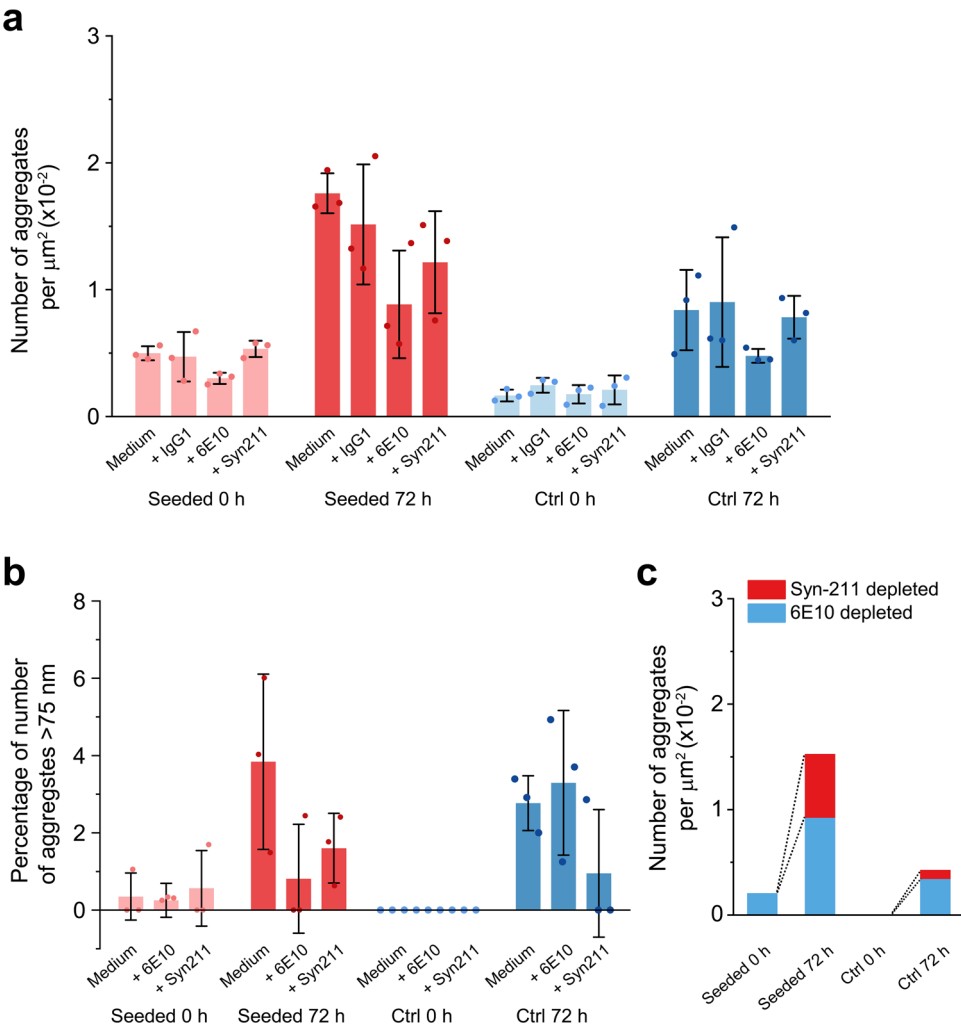

**Fig. 4 Seeded cells secrete high levels of Aβ and α-syn aggregates.** Immunodepletion of Aβ and α-syn from culture medium was carried out using 6E10 or Syn211 antibody. The immunodepleted media were then imaged with AD-PAINT. **a** Number of aggregates per μm² after immunodepletion. **b** Percentage of aggregates >75 nm in the detected population. **c** Numbers of aggregates removed by depletion with antibodies against Aβ and α-syn at defined time points. The numbers were calculated using the mean values of medium only controls and immunodepleted media as shown in **a**. Secreted aggregate of both Aβ and α-syn accumulate over time.

extracellular environment, and cell division diluting the quantity of aggregates inside cells. Strikingly, both fibril growth and fragmentation were faster in cells than in the test tube. Proteasome-mediated disassembly of fibrils accelerates fragmentation by 1–3 orders of magnitude ($7 \times 10^{-10}$–$7 \times 10^{-7}\,\mathrm{s}^{-1}$ vs. $10^{-10}\,\mathrm{s}^{-1}$ in the test tube), while fibril growth rate measured here (0.01–10 s$^{-1}$) is faster than in vitro ($8.6 \times 10^{-4}$–$1.7 \times 10^{-3}\,\mathrm{s}^{-1}$ at the 10–20 μM concentration of α-syn in SH-SY5Y cells[41]) by 1–4 orders of magnitude. Since proteasomes are thought to maintain protein quality control in cells by a refolding/degradation mechanism, this finding suggests that proteasomes may play an opposite role during severe aggregated protein stress by fragmenting and producing more seeds in cells. We did not investigate why fibril growth is faster in cells, but enrichment of α-syn at lipid membranes or in subcellular compartments are possible explanations[42,43].

Many of our images of seeded cells show aligned patterns of α-syn aggregates (mostly along the plasma membranes), with varying sizes and patterns between cells. These patterns consist of many single aggregates that can be distinguished in the zoom-in images as shown in Fig. 1b. This phenomenon can be seen at later times following, from 24 to 72 h when α-syn aggregates accumulate at higher levels. We did not carry out an in-depth study in

this regard, but these observations are consistent with the previous finding that in the presence of a-syn fibrillar seeds, the accumulation of exogenous αS fibrils takes place inside the cytoplasm within close proximity of the nucleus at early times[44].

Another key finding is that in the absence of transfection agents, ~10⁸ encounters with PFFs were required to seed a cell, despite the fact that primary neurons efficiently internalize α-syn seeds in just 1 h[45,46]. Given that internalized aggregates are localized in the endosomal-lysosomal system for hours after uptake[47], these aggregates presumably fail to seed cells because the vast majority are sequestered or degraded before they can enter the cytosol. In contrast, seeding by prion protein (PrP) happens at the cell surface within minutes of exposure[48], suggesting that PrP evades cytosolic ALP more efficiently that α-syn. If seeding only occurs after ALP is overwhelmed, then this explains the low seeding probability observed. This seeding probability is likely to decrease even further with decreasing seed concentrations, because the cells have fewer aggregates to deal with at the same time. This is recently observed in the acute dose dependence of seeding with iPSC-derived human neurons at sub-micromolar concentrations of α-syn seeds[49].

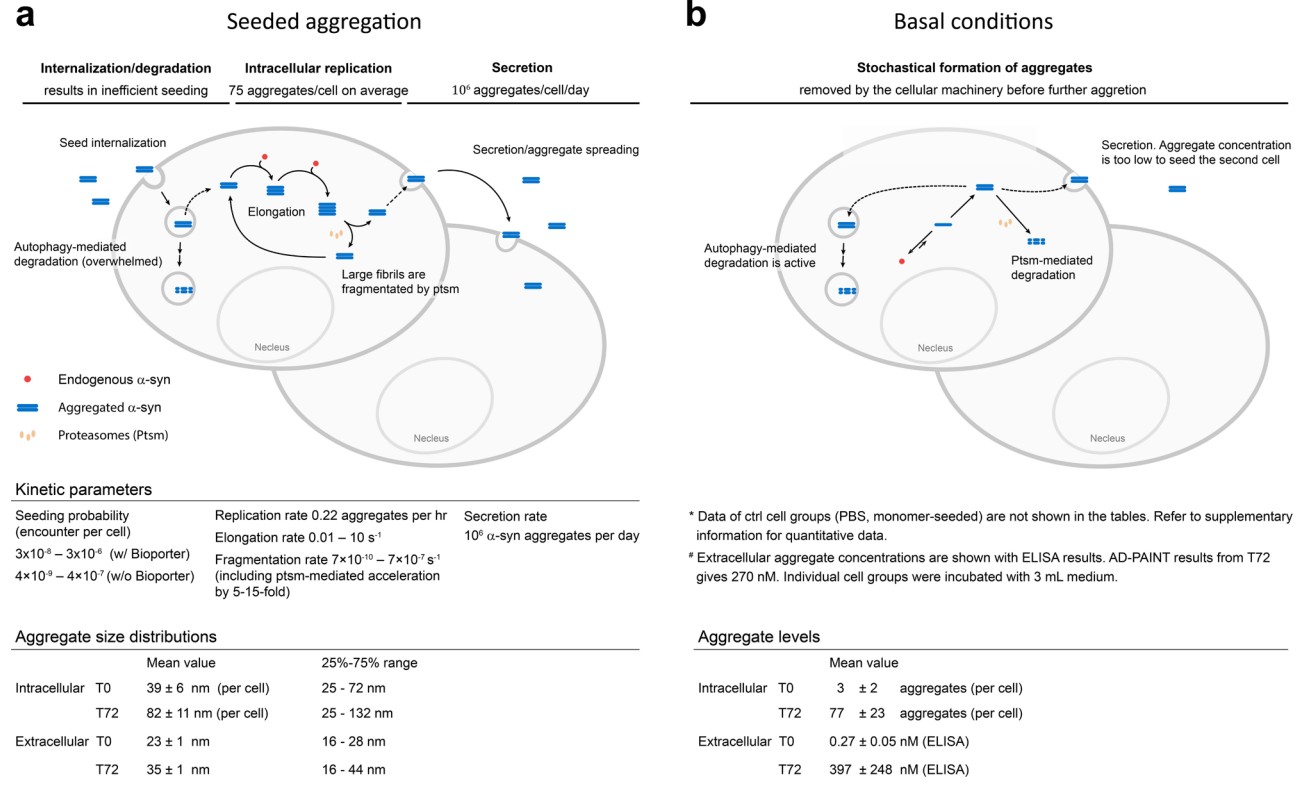

**Fig. 5 Schematic summary of α-syn seeded aggregation in SH-SY5Y cells.** The schematic summarizes the cellular responses after α-syn seeds are internalized measured in this study. In **a**, the cells can only be seeded by the fibrillar form of α-syn aggregates after sonication, not by monomeric form, in an inefficient way (shown as low seeding probability). The α-syn seeds that escape from autophagy-dependent degradation effectively replicate in cells by recruiting endogenous α-syn monomers. Proteasomes, which are responsible for protein degradation and quality control in cells, facilitate the replication process by accelerating fibril fragmentation. The resulting α-syn aggregates are secreted by the cells and accumulate in the cytosol. **b** In contrast under basal conditions, aggregates that form in the cytosol are either degraded or secreted by the cell; hence there is no accumulation of aggregates.

The current work observes seeded aggregation with endogenous α-syn without dye labeling, so that the physiological structure and behavior of fibrils/monomers are maintained as closely as possible. Seeded α-syn aggregation in cells has previously been reported using external dye-labeled fibrils and SR imaging[50]. Given that, compared to our work, the study used larger seeds and 100× lower seed concentrations without mediators, it is likely that the majority of the seeds stayed in the endosomal-lysosomal system as seen here, but lysosomal degradation was less likely to be overwhelmed by internalized seeds. In contrast, the current work sees overwhelmed ALP, which allows more seeds to escape to the cytosol, resulting in larger numbers of smaller α-syn aggregates. The aggregates in the cytosol can be further fragmented by the proteasome, resulting in even smaller aggregates. In addition, the dye-labeled fibril seeds have a high proportion of unlabeled monomers, which can generate unlabeled seeds, especially small aggregates, leading to undetectable heterofibrils and hence a bias to the detection of larger aggregates. Because our experiments target unlabeled endogenous proteins, we are able to measure the increasing proportion of aggregate-positive cells which might otherwise be rendered undetectable by dilution of extrinsically labeled monomers. This allows us to further study the time-dependent evolution of extracellular α-syn aggregates secreted by these cells.

Our data in Fig. 3 show that the number of aggregates secreted by both seeded and non-seeded cells reaches a plateau, and that these aggregates are similar in size and shape. The main difference is that the plateau is higher for the seeded cells. Our hypothesis is that when α-syn aggregation stochastically occurs in

cells, under normal physiological conditions, the aggregated assemblies are either degraded or secreted to the extracellular environment to prevent intracellular accumulation of the aggregates. Therefore, a fraction of the resulting aggregates can be detected in the media over time. The secretion level is limited for non-seeded cells and not sufficient to initiate aggregation in other cells with time.

From the study of the culture medium, Aβ aggregates are seen to be continuously secreted by SH-SY5Y cells regardless of seeding. Interestingly, seeded cells secreted higher levels of both Aβ and α-syn aggregates than control cells. We have previously seen that α-syn can spontaneously aggregate into small oligomers at a concentration as low as 0.5 μM[12], so it is presumed that this process is also taking place in the cell. We also hypothesize that disrupted protein homeostasis, due to α-syn seeding, leads to increased aggregation of both Aβ and α-syn. Soluble aggregates of both Aβ and α-syn may therefore contribute to the development of PD and changes in the proportion of a-syn aggregates may be a potential biomarker for the disease. Importantly, since most of the aggregates are secreted and far fewer aggregates accumulate inside cells, aggregate secretion appears to be a major protective mechanism against disruption of protein homeostasis. Consistent with this observation, α-syn aggregates have been shown to be secreted by axons of primary cortical neurons[51], and elevated levels of α-syn aggregates are present in exosomes in PD patients' cerebrospinal fluid[52]. In our work, the secreted aggregates are on average 35 nm in size, and only 1% of Aβ-depleted aggregates at T72 are >75 nm (Fig. 4b). However, since α-syn aggregates <75 nm were found to lack seed-competency[53], these

α-syn aggregates are unlikely to seed aggregation, but could cause neuroinflammation[54]. It would be of interest to extend these studies to neuronal models in future work.

In summary, we find that fibrillar α-syn aggregates do not efficiently seed aggregation in cells, but do disrupt protein homeostasis, leading to greatly increased secretion of nanoscopic aggregates of both Aβ and α-syn. Secretion of small aggregates appears to be a major mechanism used by cells to prevent disruption of protein homeostasis.

## Methods

**Preparation of α-syn seeds**. Expression and purification of full-length wild-type α-syn were adapted from a previously established protocol[55]. Generally, α-syn was expressed in *Escherichia coli* using plasmid pT7-7 (courtesy of the Lansbur group, Harvard Medical School, Cambridge, MA). After 20-s heat-shock at 42 °C, transformed BL21 competent cells were grown in lysogeny broth (LB) medium in the presence of 100 μg/ml ampicillin. Cells were then transferred to 1 l of LB, IPTG-induced at the final concentration of 1 mM, and cultured for 4 h at 37 °C. After expression, cells were collected by centrifugation (Beckman, Avanti J25 centrifuge with a JA-20 rotor) at 5000 rpm at 4 °C for 45 min. The pellet was resuspended with the lysis buffer [10 mM Tris-HCl (pH 8.0) supplemented with 1 mM EDTA and 1x protease inhibitor cocktail (Thermo Scientific, Pierce Protease Inhibitor Mini Tablets, Cat. A32953). and lysed by sonication (Fisherbrand, Model 705 Sonic Dismembrator). After centrifugation at 13,000 rpm at 4 °C for 30 min, the supernatant was collected, boiled for 20 min at 80–95 °C, and centrifuged at 13,500 rpm at 4 °C for 30 min. Then, streptomycin sulfate was added to the supernatant to a final concentration of 10 mg/ml and the mixture was stirred for 15 min at 4 °C, then centrifuged again at 13,500 rpm at 4 °C for 30 min. α-syn was precipitated by adding ammonium sulfate to a final concentration of 0.36 g/ml and then stirred for 30 min at 4 °C. After centrifugation at 13,500 rpm at 4 °C for 30 min, the pellet was collected and resuspended in 25 mM Tris-HCl (pH 7.7). The solution was dialyzed overnight with 3.5k MWCO membranes (Spectrum™ Spectra/Por™ 3 RC Dialysis Membrane Tubing, Cat. 10142634) in 4-l dialysis buffer of 25 mM Tris-HCl (pH 7.7). Ion-exchange chromatography was carried out with an HQ/M-column (Q Sepharose High Performance from Cytiva) on an Applied Biosystems BIOCAD workstation. α-syn was eluted roughly at the level of 300 mM NaCl with a salt gradient from 0 to 600 mM NaCl. The protein solution was dialyzed overnight against the appropriate buffer until use. The purity of α-syn was judged by SDS-PAGE, electrospray ionization mass spectrometry, and analytical gel-filtration. Protein concentration was estimated from the absorbance at 275 nm using an extinction coefficient of 5,600 $M^{-1}$ $cm^{-1}$.

The aggregation reaction was carried out in a 1.5 ml microcentrifuge tube containing phosphate-buffered saline (PBS) with 0.1% $NaN_3$ at a starting concentration of 70 μM and a volume of 300 μl. After 14 days of 37 °C incubation in the dark with 200 rpm shaking in an orbital incubator (Innova 43, New Brunswick Scientific), PFF seeds were generated as described[11]. Briefly, the fibrils were suspended and sonicated for 10 min using Sonorex Super RK-52 (Bandelin, Germany) with an effective power of 60 W.

**Cell culture, seed transduction, and treatments**. SH-SY5Y cells were maintained in Dulbecco's modified Eagle's medium (DMEM, Thermo Fisher, Cat. 11995065) containing 10% fetal bovine serum (FBS, US sourced HyClone characterized, GE) and 1% penicillin-streptomycin (Thermo Fisher, Cat. 15140122) in a humidified 37 °C/5% $CO_2$ environment. Before seed transduction, cells were transferred onto a round borosilicate coverslip (0.13 mm thickness, Ø = 20 mm) in a 6-well tissue culture plate (Greiner CELLSTAR, Cat. M8562), allowed to reach ~50% confluence. The cells were then fully rinsed with warm PBS (Thermo Fisher, Cat. 10010023) and incubated with serum-free Opti-MEM (Thermo Fisher, Cat. 31985062) at 37 °C for 1 h. To introduce α-syn seeds into the cells, the cationic-liposomal transduction reagent Bioporter (Sigma-Aldrich, Cat. BPQ24) was used according to the manufacturer's instructions. For each well in the plate, one reaction tube of lyophilized Bioporter reagent was resuspended with PBS and gently mixed with 35 μg of α-syn seeds, then placed on the benchtop at room temperature. Protein-reagent complexes could form in 5 min. The mixture was diluted with 1 ml Opti-MEM to a final protein concentration of 2.5 μM (in monomer equivalents); then replacing the cell medium with the mixture. After 4 h incubation at 37 °C, the cells were thoroughly rinsed three times with PBS to remove residual seeds in the cell medium, then cultured with DMEM (Thermo Fisher, Cat. 21063045) supplemented with 10% FBS and 1% penicillin-streptomycin at 37 °C until collection.

BTZ (Item No. 10008822), MG132 (Item No. 10012628), CFZ (Item No. 17554) were purchased from Cayman Chemical (USA); 3-MA (Cat. M9281), BafA1 (Cat. 5084090001), and rapamycin (Cat. 553210) were purchased from Sigma-Aldrich. Each compound was dissolved in dimethyl sulfoxide (DMSO) with at least 1000× higher than its working concentration and stored at −80 °C until use. For proteasomal inhibition experiments, cells were treated with BTZ, MG132, or CTZ at a working concentration of 5, 1, and 1 μM, respectively, at T0. For autophagy modulation experiments, cells were treated with 3-MA, BafA1, or rapamycin at

working concentrations of 2.5 mM, 0.4 μM, and 60 nM, respectively, at T0. After 24 h incubation at 37 °C, the cells were fixed and then imaged.

TNF-α (Cat. H8916) and NS398 (Cat. N194) were purchased from Sigma-Aldrich, dissolved with DMSO, and stored at −80 °C. Unseeded cells were treated with either high (10 ng/ml) or low (50 pg/ml) dose of TNF-α in the presence or absence of 100 μM NS398. After 48 h incubation at 37 °C, the cells were fixed and then imaged.

**Cell viability assay**. Cell viability was measured using an LDH assay kit (Abcam, Cat. ab102526) following the manufacturer's protocol. Briefly, 25 μl of cell medium was removed from the cell culture and mixed with 25 μl of assay buffer. Fifty microliters of substrate mix was added into each reaction for LDH activity detection. After 30 min incubation at 37 °C, the absorbance at 450 nm was measured on a plate reader. The viability of unseeded control cells (cell density ~5 × 10^4 per $cm^2$) at time = 0 h was normalized as 100%.

**Immunodepletion**. To immunodeplete Aβ or α-syn in culture medium, 6E10 (anti-Aβ epitope 1-16; Abcam, Cat. ab80627) and Syn211 (anti-α-syn epitope 121-125; Biolegend, Cat. 803004) antibodies were used. IgG1 control antibody (clone COLIS69A; Kingfisher Biotech, Cat. WS0799M-100) was used as a negative control. To bind the antibodies to magnetic beads, 50 μl of Dynabead-protein G (Invitrogen, Cat. 10003D) was transferred to a 1.5 ml centrifugal tube. After removing the preserving solution, 200 μl of antibodies were separately diluted with PBS to 25 μg/ml and thoroughly mixed with the beads for 1 h at room temperature. After removing the excess antibody solution, the beads were washed with PBST (0.1% tween-20) for 10 min and this was repeated twice. The antibody-bound beads were incubated with 75 μl of medium samples for 30 min at room temperature. The solution was then separated from the beads and transferred onto a multi-well chambered coverslip for AD-PAINT.

The total population detected in the culture medium comprises α-syn, Aβ, and the baseline signal. The baseline signal derives either from the non-specific binding of the aptamer with sample components or glass surface, or from stochastic interaction between the imaging strand (IS) and the surface. Note that the latter interaction is minimized after surface blocking and is typically seen ≤1 cluster per image on average. To acquire the fractions of α-syn and Aβ aggregates in total population, a set of simultaneous equations consisting of the variables of α-syn, Aβ, and the baseline signal was solved using the mean values of medium only controls and immunodepleted media as shown in Fig. 4a.

**α-syn ELISA**. Culture medium was collected at defined time points after seeding. The samples were flash frozen and stored at −80 °C until use. Sandwich ELISAs were performed using the Human Alpha-synuclein ELISA Kit (Abcam, Cambridge, UK, Cat. ab260052) and following the manufacturer's instruction.

**Transmission electron microscope**. To prepare TEM samples, protein solutions were applied on a carbon-coated 400-mesh copper grid and fixed for 1 min. The excess solution was removed from the surface by rinsing three times with distilled water. Negative staining was carried out using 2% (v/v) uranyl acetate for 1 min, followed by another three-time rinsing with distilled water. After air drying, the grid sample was loaded onto a Thermo Scientific Talos F200X G2 scanning transmission electron microscope. Image acquisition was carried out at 200 kV.

**Oligonucleotides**. All oligonucleotides (synthesized on the 0.2 or 1.0 μmol scale) were purchased from ATDBio (Southampton, UK) and purified by HPLC, except fluorophore-labeled IS1, which was purified by double HPLC. Lyophilized thiolated docking strand (DS) (Thiol-DS1) received from the supplier was dissolved in 100 μL 1× PBS (pH 8.5, containing 50 mM DTT, dithiothreitol) and incubated for 30 min to reduce any residual disulfides. This solution was then diluted with water (400 μl) and extracted with ethyl acetate (3 × 500 μl). The DTT-treated oligonucleotide was then desalted using a NAP-5 column (GE Healthcare, Cat. No. 17085301) and its concentration calculated from $A_{260}$ (Nanodrop 2000, Thermo Fisher Scientific). All other lyophilized oligonucleotides were dissolved in deionized water to concentrations of 20–1000 μM as confirmed by $A_{260}$ and aliquots stored at −20 °C.

**Antibody labeling**. In this study, we used two different batches of MJFR-14-6-4-2 antibody, BSA and Azide free (Abcam, Cambridge, UK, Cat. ab214033) for DNA-PAINT: one based on cross-coupling Thiol-DS1 (Table S1) to random lysine residues[56] exactly as in our previous work[31] (antibody lot number GR310602-4, used for data in Figs. 1 and S2, S5, and S11) and the other based on selective conjugation to carbohydrates of the Fc region (antibody lot numbers GR323266456-2 and GR3266456-3 combined in one reaction, used for all other data). In this method, the carbohydrates were first functionalized using a SiteClick™ Antibody Azido Modification Kit (Invitrogen, Cat. No. S20026) according to the manufacturer's instructions. Briefly, ~200 μg of antibody was concentrated to 5.4 mg/ml in antibody prep buffer (as supplied in the kit), treated overnight with β-galactosidase for 16 h at 37 °C, then coupled to UDP-GalNAz using β-1,4-galactosyltransferase for 16 h at 30 °C. Following purification on Amicon spin filter (50

kDa MWCO), the antibody (60 µg at 3.7 mg/ml) was then coupled to ten molar equivalents of dibenzocyclooctyne DS1 (4.4 nmol) using copper-free click reaction in 1× PBS supplemented with 50 mM NaCl. After incubation for 16 h, the excess oligonucleotide was removed using an Amicon spin filter (50 kDa MWCO) and the concentration of antibody and degree of labeling (~3–4 DS per antibody) measured using $A_{260}$ and $A_{280}$. The purity and labeling efficiency were further confirmed using SDS-PAGE under reducing conditions (Fig. S3).

**Super-resolution imaging**. To prepare cell samples for imaging, cells were rinsed with cold PBS three times, then fixed with paraformaldehyde (4% w/v) for 15 min at room temperature, followed by PBS rinsing. Cells were then permeabilized with 0.3% Triton X-100 (Sigma) in PBS for 20 min, followed by PBS rinsing. Potential non-specific binding sites were blocked by 1 h incubation with a blocking buffer consisting of 3% BSA (Fisher Scientific, Cat. BP9704-100), 10% sheared salmon sperm DNA (UltraPure, Thermo Fisher, Cat. 15632-011), and 0.1% $NaN_3$ in PBS. Culture medium was collected, flash frozen, and stored at −80 °C until use.

To super-resolve intracellular α-syn aggregates, antibody-DNA-PAINT was employed in a combination of MJFR-14-6-4-2 antibody (Abcam, Cat. ab214033). The DS-labeled antibody was diluted with the blocking buffer to a final concentration of 0.67 nM. This DS concentration, in combination with the use of the current permeabilization/blocking method, is optimized for cell imaging and shows very few non-specific binding clusters (Fig. S5) as mentioned in our previous study[31]. The cells were incubated with the antibody solution overnight at 4 °C. Before imaging, the solution was replaced with an imaging buffer (5 nM Cy3b-labeled IS in the blocking buffer) in the presence of 3000× diluted Hoechst 33342 (Sigma-Aldrich, Cat. H3570). Imaging was performed using near-total internal reflection fluorescence (TIRF) illumination at 561 and 405 nm, where illumination light is propagating at a shallow angle to the focal plane.

To super-resolve extracellular aggregates in the culture medium, aptamer DNA-PAINT (AD-PAINT)[31] was employed on a glass surface, due to excess non-specific adsorption from the antibody onto glass coverslips that results in false-positive signals. A round borosilicate coverslip (0.13 mm thickness, Ø = 50 mm) was cleaned using a plasma cleaner (PDC-002, Harrick Plasma) with argon gas for 1 h. A multi-well chambered silicon gasket (diameter × thickness = 3 mm × 1 mm, CultureWell, Grace Bio-Labs, Cat. GBL103350) was cut in half and affixed to the round coverslip. The glass surface was then coated with 1 mg/ml L-aspartic acid (Sigma, Cat. A8949) for 30 min, then rinsed once with PBS. Cell medium (10 µl) was loaded into a well and incubated for 30 min before removal. The DS-labeled aptamer was diluted with the blocking buffer to a final concentration of 100 nM, then incubated with each well. Before imaging, the solution was replaced with the same imaging buffer. Imaging was then performed using TIRF illumination at 561 nm. To correct the spatial drift of images in x–y dimensions during acquisitions, fiducial markers (TetraSpeck 0.1-µm fluorescent microspheres, Thermo Fisher Scientific, Cat. T7279) with 1000× dilution was applied during DS-antibody or DS-aptamer incubation. All reagents for SR imaging were 0.02-µm filtered with Anotop25 (Whatman, GE) before use.

**Instrumentation**. SR imaging was performed using a home-built TIRF microscopy setup. Two lasers operating at 405 nm (LaserBoxx, Oxxius, France) and 561 nm (Cobalt Jive, Cobalt, Sweden) were used for Hoechst and Cy3B illumination, respectively. The lasers were aligned to the optical axis of a high numerical aperture, oil-immersion, ×100 objective (CFI Apochromat TIRF 100XC Oil, Olympus, Japan, NA = 1.49) mounted on an inverted Ti-E Eclipse microscope (Nikon, Japan). The microscope was fitted with a Nikon perfect focus system that auto-corrects the drift along the z-axis during imaging. The laser power was attenuated by neutral density filters before the beam passed through the respective excitation filters (FF01-417/60-25 for Hoechst and FF01-561/14-25 for Cy3B, Semrock, USA) and circularly polarized by quarter-wave plates. To combine the beams, the lasers passed through a dichroic mirror (FF458-Di02-25 × 36, Semrock, USA) and directed into the objective. Total internal reflection was achieved by focusing the laser beam at the back focal plane of the objective. For typical TIRF imaging, the emergent beam at the sample interface was near-collimated and incident at an angle greater than the critical angle $\theta_c$ (~67°). The emitted fluorescence was collected through the same objective, separated from the excitation light by a dichroic mirror (Di01-R405/488/561/635, Semrock, USA), subsequently passed through appropriate filters (BLP01-488R-25 for Hoechst dye and LP02-568RS-25 for Cy3b, Semrock, USA), and finally recorded by an EMCCD camera (Evolve 512, Photometrics, USA) with an electron multiplication gain of 250. The image stacks were automatically acquired using the open-source microscopy platform Micromanager[57] with a custom-written script. The acquisition typically lasted for 4000 frames for 561 nm channel and 100 frames for 405 nm and brightfield channels, with an exposure time of 50 ms. For antibody-DNA-PAINT and AD-PAINT, separate instruments with the same configuration were used. The image size is 512 × 512 with a pixel size of 98.6 nm (antibody-DNA-PAINT) and 107 nm (AD-PAINT). The localization precisions of antibody-DNA-PAINT and AD-PAINT are determined to be 12.2 ± 0.5 nm and 19.0 ± 6.6 nm, respectively, in this study. In single-molecule localization microscopy (SMLM), localization precision σ, or standard deviation of localizations, of a point source is limited by its own point

spread function (PSF) and can be approximated by the equation below[58]:

$$\sigma \approx \frac{\sigma_{PSF}}{\sqrt{N_{signal}}} \tag{1}$$

where $\sigma_{PSF}$ is the full width at half maximum of the PSF and $N_{signal}$ is the photon count.

**Image analysis**. The raw image stacks were passed through a custom analysis workflow script that was programmed in Jython 2.7 and Python 3.5. The analysis is divided into three parts: (1) localization fitting, where raw fluorescence signals in the image stacks are fitted with localizations with their information in a tabulated form; (2) cluster analysis on localizations based on their spatial and temporal information to identify individual clusters; and (3) characterization of the morphology of individual clusters.

We defined a localization as a 2D Gaussian fit of a fluorescence signal in one image frame. The localizations were determined using Fiji[59] with the "Peak Fit" function in GDSC Single-Molecule Light Microscopy (GDSC-SMLM) package: (http://www.sussex.ac.uk/gdsc/intranet/microscopy/UserSupport/AnalysisProtocol/imagej/gdsc_plugins/).

The adjustable parameters "signal strength" and "precision" were set to 40 and 25 nm, respectively. These values are associated with the selection criteria of the 2D Gaussian fit and were set by visual inspection of the rendered localization images. The fiducial markers (TetraSpeck 0.1-µm fluorescent beads) in images were identified as the localizations that lasted more than 100 frames at the same location. Positions of all localizations were then corrected with the fiducial drift using the "Drift Calculator" plugin in GDSC-SMLM package. The corrected results were saved in a file for subsequent analysis that contains information about time frames, x–y coordinates, precision, signal strength, and background strength.

The localization data were first filter-passed to remove fiducial signals; then they were transformed to grouped bursts by temporal grouping to identify DS bound to targets and to decouple the effects of binding kinetics. Temporal grouping was achieved by: (1) using Density-Based Spatial Clustering of Applications with Noise (DBSCAN) clustering function (scikit-learn) on the spatial domain (x–y coordinates) of the localizations, with a detection radius (epsilon) of 30 nm and minimum localization threshold of 3 to identify docking strand binding sites; (2) using DBSCAN again on the temporal domain (frame number) of each DS site, with an epsilon of 2500 ms and minimal frame number of 1 to recognize individual bursts and remove single-frame localizations. These single-frame localizations are removed because non-specific binding events are typically characterized by short transient signals that last less than the exposure time of the camera. (3) Passing the burst information through the final DBSCAN analysis on the spatial domain using an epsilon of 60 nm and a minimal burst number of 2 to identify clusters. The clusters identified are considered to be super-resolved protein aggregates in this study.

The lengths of individual clusters were identified by: (1) scaling up the spatial domain by eight times and rounding the coordinates to integers; (2) closing the morphological space between bursts using the scikit-image "closing" function; (3) skeletonizing the closed shape to a width of one pixel using the scikit-image "skeletonize_3d" function; (4) calculating the lengths by traversing the skeleton and recording the 8-connectivity distances. Finally, clusters were loaded into the "Results Manager" plugin in Fiji/GDSC-SMLM and rendered into super-resolved images with labels using a pixel size of 13 nm, smaller than the mean localization precision.

**Statistics and reproducibility**. Experimental data are represented as mean ± SD with three biological replicates. The data were plotted with superimposed scatter points to demonstrate variance within (small circles) and among (large circles) datasets. Biological replicates are termed "N"; number of cells analyzed is termed "n" in the main text. Data were assessed for normality using the Shapiro–Wilk test and the Kolmogorov–Smirnov test. Statistical analysis was performed using one-way ANOVA (Figs. 1 and 2) or two-way ANOVA (Fig. 3), followed by Tukey's multiple comparison test with a significance threshold of $p = 0.05$. The software OriginPro 2017 (OriginLab) was employed for the statistical analysis.

**Kinetic analysis of the aggregate formation**. We used a minimal model of self-replication with a limiting maximal concentration. This is described by the differential equation:

$$\frac{df}{dt} = kf(t)\bigl(1 - f(t)\bigr) \tag{2}$$

which can be solved to give:

$$f(t) = \frac{f_0 \exp(kt)}{1 - f_0 + f_0 \exp(kt)} \tag{3}$$

where $f(t)$ is the fraction of the number of aggregates at time $t$ relative to the maximum number of aggregates, $f_0$ is its initial value at time = 0 h, and $k$ is the replication rate. In our SR measurements, the information was collected from each cell as a function of time. The median values of each time point were fitted to $P_{max}f(t)$, where $P_{max}$ denotes the maximal number of aggregates.

In the kinetic fitting in Fig. 1e, there were three free parameters, $f_0$, $k$, and $P_{max}$. We analyzed the data consisting of individual measurements using Bayesian inference, with lognormally distributed noise, flat priors for $k$ and $P_{max}$, and a $1/f_0$ prior for the initial fraction. Errors given in the main text are the 95% credible intervals for the marginalized distributions for $k$. Fitting the means (of the medians for individual biological replicates) of the data, we obtained the best fit for $k = 0.22\,h^{-1}$, with a possible range between 0.14 and 0.41 $h^{-1}$ at 95% credible intervals (Table S2). The accuracy is within a factor of 3, which is reasonable for a reaction rate.

In the analysis of the proteasomal inhibition experiments from Fig. 2, it is assumed that the presence of inhibitors does not alter either the initial fraction, $f_0$, or the final number of aggregates at equilibrium, $P_{max}$. Thus, we used only one free parameter, the replication rate $k$, and fixed $f_0$ and $P_{max}$ to the values determined in the absence of proteasome inhibitors. Note, as there is only one-time point measurement in the presence of proteasome inhibitor, fitting the mean number of aggregates in this case is equivalent to solving Eq. (3) for $k$.

The effective rate of replication, $k$, is decomposed into the rates of growth ($k_g$) and fragmentation ($k_f$) by using the mean aggregate length. The replication rate is the geometric mean of the rates of growth and fragmentation[11]:

$$k = \sqrt{k_g k_f} \tag{4}$$

whereas the average size, $\mu$, (in monomer equivalents) is given by the square root of their ratio:

$$\mu = \sqrt{k_g/k_f} \tag{5}$$

Thus, with $k$ solved, determining the mean aggregate length allows disentangling of the rates for growth and fragmentation. To determine the mean aggregate length, we use a conservative estimate with a possible range to approach the number of monomers incorporated in an aggregate 100 nm in length, which is 400–40,000. With $k = 0.22\,h^{-1}$, the $k_g$ and $k_f$ are derived to be $0.01$–$10\,s^{-1}$ and $7 \times 10^{-10}$–$7 \times 10^{-7}\,s^{-1}$, respectively.

**Estimation for seeding probability.** In our seeding experiments, the concentration of fibrillar seeds is 2.5 μM monomer equivalents. These seeds are about 100 nm in length containing 400–40,000 monomers (conservative estimate), giving a seed concentration of ~0.06–6 nM. In the culture medium of 3 ml, there are therefore about $10^{11}$–$10^{13}$ seeds. Since there are $5 \times 10^4$ cells per cm² during seeding with an area of wells of 9.6 cm², this is roughly $2 \times 10^7$–$2 \times 10^9$ seeds per cell, hence a substantial excess of seeds over cells. This large excess of seeds over cells is the normal situation in most cell seeding experiments. The number of aggregates that encounter a cell will depend on the aggregate concentration and exposure time. Successful seeding requires an aggregate to enter the cell and escape the protein clearance machinery in order to seed aggregation. We can calculate the probability that encounters of a seed with a cell leads to seeding by estimating the number of encounters a seed makes with a cell in the 4 h incubation period and the measurement of the % of cells that as a result are seeded, defined to have >10 aggregates at T24.

The diffusion coefficient $D$ of an aggregate is given by:

$$D = \frac{kT}{6\pi\eta R_H} \tag{6}$$

where $k$ is Boltzmann constant, $T$ is absolute temperature, $\eta$ is viscosity, and $R_H$ is the hydrodynamic radius.

During seeding, only a fraction of the aggregates present in the solution can encounter cells in a given time. The local aggregate concentration $c$ is governed by the diffusion equation:

$$D\nabla^2 c = \frac{\partial c}{\partial t} \tag{7}$$

Consider a spherical cell of radius $a$ in the medium. In the steady state, the flux $J$ of aggregates to the cell (i.e., the number of aggregates that encounters the cell per unit time) is given by:

$$J = 4\pi a D c \tag{8}$$

where $c_\infty$ is the aggregate concentration far from the cell (number per unit volume), which is assumed to be a constant. The unit of $J$ is in molecules per second.

The effective $R_H$ is ~50 nm and the literature viscosity[60] of DMEM + 10% FBS is $0.94 \times 10^{-3}$ Pa s. Therefore, at 37 °C, the diffusion coefficient of aggregates is estimated to be $4.8 \times 10^{-12}\,m^2\,s^{-1}$, according to Eq. (6). For a SH-SY5Y cell with a radius of 10 μm, $J$ is therefore 200–20,000 molecules per second. Hence, the number of seeds that encounter each cell during 4 h seeding is $3 \times 10^6$–$3 \times 10^8$.

The percentages at T24 (Fig. S6) was 12.5% (in the absence of Bioporter) after 4 h seeding. Therefore, the seeding probability without Bioporter is approximately $4 \times 10^{-9}$–$4 \times 10^{-7}$, so of order of magnitude ~$10^{-8}$.

There is not much available data for comparison of the seeding probability derived here. However, a recent study[61] carried out seeding with a primary culture of murine hippocampal neurons using sonicated α-syn fibrils at concentrations from 4 to 500 nM over 14 days and observed the % cells with phosphorylated α-syn between 10 and 50%. This gives an estimated seeding probability of a similar order of magnitude, $10^{-8}$.

**Reporting summary**. Further information on research design is available in the Nature Research Reporting Summary linked to this article.

## Data availability

Source data for Figs. 1–4 are provided with the paper as Supplementary Data 1. All other data that support the findings of this study are available from the corresponding author upon request.

## Code availability

The original codes that support the findings of this study are available in a DOI-minting repositor Zenodo with the identifier (doi: 10.5281/zenodo.4651484). The version used is v3.5 (most updated). All the variables used for analysis are described in the section of Image analysis.

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

## Acknowledgements

The authors wish to thank Dr. Yu Ye for providing the cell-line, Swapan Preet for expression and purification of α-synuclein, Dr. Heather Greer of the electron microscopy facility of the Department of Chemistry for assistance with TEM imaging, and Dr. Joel Watts at the University of Toronto for helpful and stimulating discussions. This work was supported by the UK Dementia Research Institute which receives its funding from DRI Ltd., funded by the UK Medical Research Council, Alzheimer's Society and Alzheimer's Research UK, and by the European Research Council Grant Number 669237 and the Royal Society.

## Author contributions

J.C.S. designed and performed experiments, analyzed data, and wrote the manuscript. E.H. performed the AD-PAINT experiments and analyzed the data, and established the codes for image analysis. G.M. performed the kinetic analysis for the experimental data and interpreted the results. R.T.R. provided DNA-labeled MJFR antibody and wrote the manuscript. M.G.S. and D.K. conceptualized and supervised the study and wrote the manuscript. All authors discussed the data and contributed to the writing of the manuscript.

## Competing interests

The authors declare no competing interests.
