## [Peer Review File · Communications Biology]

Reviewers' comments:

Reviewer #1 (Remarks to the Author):

The manuscript has the agenda to shed light on the nature of a-syn aggregation and seeding, which is not fully understood. Therefore the topic is of high interest and results could potentially give a better understanding of the seeding and cell to cell transfer of a-syn. I recommend that the study is considered for publication if additional experiments are performed.

1. When this is a manuscript that attempts to improve our basic understanding of a-syn cell to cell seeding then it is a bit strange that differentiated neurons are never used in this study. I would appreciate if primary neurons were also analysed. I recommend that such a study is carried out. It could be carried out with seeding in a similar fashion as performed by group of V. Lee (Nat Protoc. 2014 Sep; 9(9): 2135–2146.)

2. Could the authors comment on why there is no increase in aggregation from 24 to 72 hours? If

3. Could the authors comment on why there is a dramatic increase in number of a-syn aggregates even without stimulation with a-syn seeds in the controls of figure 4

4. When the cells were "transfected" with a-syn seeds complexed with bioporter was the control cells then given bioporter alone or bioporter plus control protein? Bioporter plus control protein that does not cause a-syn aggregation would be the most proper control

Reviewer #2 (Remarks to the Author):

This manuscript by Sang et al., reports on alpha-syn aggregation and seeding in SH-SY5Y cells, via the use of a specific type of super resolution microscopy, based on the so-called antibody PAINT. While the topic is of interest, in my view the paper lacks proper experimental support of the conclusions drawn and the novelty required.

While alpha-synuclein seeding is the main focus of the manuscript, the authors fail to show proper super-resolved images of fibrils and aggregates and distinguish between the endogenous alpha-syn protein and the fibrils which are exogenously added to the cells.

Moreover, prior work and publications on the topic are not referenced or mentioned, more specifically the work of Pinotsi et al. in PNAS, 2016 113 (14). There, the authors performed dual-color super-resolution microscopy to distinguish between the added fibrils and the induced aggregation of the endogenous alpha-syn both in SH-SY5Y cells but also in dopaminergic neurons.

This work should be discussed and potential differences that may lead to different findings should be suggested.

In its current form, I do not support publication of this paper in Communications Biology.

In more detail, here are my comments and suggestions for improvement:

1. The paper reports on alpha-syn aggregation but there are no convincing nanoscopic images of aggregates, neither of the added fibrils nor of the induced aggregates. How do the authors distinguish between the added alpha-synuclein aggregates and the endogenous protein / induced aggregates?
2. According to the authors monomeric alpha-syn is not efficiently detected by antibody-DNA-PAINT imaging – does that hold for oligomeric too? That means that not all the effects on the aggregation and nucleation of the endogenous alpha-syn can be detected with this imaging method. The authors should also use a complementary super resolution imaging method with antibody labeling for example

to image the endogenous protein, to support their findings.

3 . How is the counting of number given that what we see in the figure are "cluster-like"? What is the unit size of each counted species?

4. Please define PFF.

5. Figure 1: Poor quality and pixelated.

Panel b: the Bright Field Images have a strange bright "shading": due to improper illumination? This panel is so small the reader cannot really "see" the aggregates and where they are located.

6. Figure 2: in this image we see a lot of aligned, "linear" aggregates, compared to the previous one. What is the reason for this?

7. Figure 3: Panel b: Why at 24h there are more aggregates than 48h for the control? There also seems to be no significant increase between 24h, 48h and 72h?

Why does the control also show increase of number of aggregates for 0, 4 and 24h?

Panel c: the control sample also has aggregates of the same size?

Panel d: it seems all aggregates in control and seeded have the same eccentricity. Why is that?

8. Figure 4: Where are the super resolution images of the aggregates mentioned?

9. In Materials and Methods: the pixel sizes of 96nm and 107 nm are large compared to the size of aggregates to be resolved.

10. Why do the authors state that micromolar is the normal seeding concentration in cells? That is true for in vitro experiments.

Reviewer #3 (Remarks to the Author):

Sang et al. report work on using superresolution microscopy (SRM) to monitor α -syn aggregation in a model neuronal cell line. It is a follow up on the previous in vitro work from the same lab, and the results shed new light on the α -syn aggregation process, particularly the role of pre-existing α -syn aggregates in seeding new aggregates, the cellular machineries potentially involved in the processing and trafficking of α -syn aggregates. The manuscript is well organized, and the data are clearly presented and convincing. It is also a nice demonstration of how SRM could help resolve biological processes involving nanoscopic structures in intact cells, which would have been difficult with conventional fluorescence microscopy or with electron microscopy. I suggest that the authors consider the following when revising the manuscript.

a) The discussions section reads nicely. To help better keep track of what we have learned and what the remaining questions are, it'd be useful to add a figure that summarizes the results so far into a working model, hopefully also incorporating answers to (and/or hypotheses for) some of the questions below;

b) What types of α -syn structures (can) actually serve as seeds for growing new α -syn aggregates? The authors mentioned two different pre-existing α -syn fibrils used for seeding, sonicated fibrils or 'soluble' ones. They used sonicated fibrils and have done some characterizations on these in the present work. What do we know about the 'soluble' aggregates in terms of size, efficiency in cellular uptake and driving new α -syn formation once in cells? Also, can the small, soluble α -syn aggregates secreted by the cells be taken up by the cells in an autocrine manner and seed formation of α -syn

aggregate formation? If so, this would yield a feed-forward process, in which case a braking mechanism would seem necessary. Correct me if I am mistaken here.

c) The product of α -syn fibril sonication is likely a mixture of aggregates in different sizes, and it would be helpful to examine whether a subset of this mixture was preferentially taken up by the cells. I presume that the aggregates observed at T=0h represent the original seeds, although some may have already been processed by the UPP and/or ALP (hence some of the data were already there)? Knowing this would be helpful to better define the seeding process.

d) When imaging low abundance species such as the α -syn aggregates at T=0h and T=4h, signals from the substrate (e.g. coverglass) could be problematic. The authors overcame this by switching from antibody-DNA-PAINT, which suffers more from nonspecific sticking to the coverglass but distinguishes α -syn aggregates better from A β to using aptamers for DNA-PAINT (AD-PAINT), which recognize both α -syn aggregates and A β but are less sticky to coverglass. Although not directly related to the biological questions, it would still be helpful for the authors to provide example raw images or videos in the supplementary material.

e) It would also be useful to include in the supplementary material schematics for imaging α -syn aggregates using either AD-PAINT or antibody-DNA-PAINT, for example how the actual dimensions of the aggregates are related to the apparent dimensions measured in the SRM images depending on the imaging strategy;

f) The legend for Figure 4 needs attention.

Reviewers' comments:

Reviewer #1 (Remarks to the Author):

The manuscript has the agenda to shed light on the nature of a-syn aggregation and seeding, which is not fully understood. Therefore the topic is of high interest and results could potentially give a better understanding of the seeding and cell to cell transfer of a-syn. I recommend that the study is considered for publication if additional experiments are performed.

1. When this is a manuscript that attempt to improve our basic understanding of a-syn cell to cell seeding then it is a bit strange that differentiated neurons are never used in this study. I would appreciate if primary neurons were also analysed. I recommend that such a study is carried out. It could be carried out with seeding in a similar fashion as performed by group of V. Lee (Nat Protoc. 2014 Sep; 9(9): 2135–2146.)

This is the first time that super-resolution imaging has been used to study the seeded aggregation of α -synuclein in unlabelled cells and to reduce the complexity of the experiments we studied a model neuronal cell line in detail. It is well beyond the scope of the current study to study primary neurons. We have now added in the discussion that it would be of interest to extend these studies to differentiated neurons in future work to address this point.

2. could the authors comment on why there is no increase in aggregation from 24 to 72 hours? If

We assume that the reviewer refers to Fig. 1b-f. The number of intracellular aggregates depends on the difference between the rate of aggregate production and removal. Aggregate growth by addition of monomers and subsequent fragmentation increases the number of aggregates, while aggregate degradation by the cellular machinery and secretion to the extracellular environment decreases the number of aggregates. The plateau of seeded aggregation can be explained by the cells reaching a steady state after an initial fast increase due to seeding. The rate of seed production is roughly balanced by the rate of seed removal due to up-regulation of the cellular machinery such as ALP, aggregate secretion into the extracellular environment, and cell division diluting the quantity of aggregates inside cells. We have added a section to explain this in the discussion.

3. could the authors comment on why there is a dramatic increase in number of a-syn aggregates even without stimulation with a-syn seeds in the controls of figure 4

Our hypothesis is that α -syn aggregates are formed stochastically and either degraded or secreted to extracellular environments under normal physiological conditions to prevent any accumulation of α -syn aggregates in the cells. The secreted α -syn aggregates accumulate in the media over time. We have added a section to explain this in the discussion.

4. when the cells were "transfected" with a-syn seeds complexed with bioporter was the control cells then given bioporter alone or bioporter plus control protein? Bioporter plus control protein that does not cause a-syn aggregation would be the most proper control

In Fig. S3 we have attached data with negative controls of PBS only and Bioporter+ α -syn monomer in PBS. Neither treatment triggers α -syn aggregation in SH-SY5Y cells. We believe that monomeric synuclein is the optimal control protein since it is the same

protein type with a different conformation from aggregates. Bioporter only (in PBS) was also tested once, with a negative result, but not shown in the data.

Reviewer #2 (Remarks to the Author):

This manuscript by Sang et al., reports on alpha-syn aggregation and seeding in SH-SY5Y cells, via the use of a specific type of super resolution microscopy, based on the so-called antibody PAINT. While the topic is of interest, in my view the paper lacks proper experimental support of the conclusions drawn and the novelty required.

While alpha-synuclein seeding is the main focus of the manuscript, the authors fail to show proper super-resolved images of fibrils and aggregates and distinguish between the endogenous alpha-syn protein and the fibrils which are exogenously added to the cells.

Moreover, prior work and publications on the topic are not referenced or mentioned, more specific the work of Pinotsi et al. in PNAS, 2016 113 (14). There, the authors performed dual-color super-resolution microscopy to distinguish between the added fibrils and the induced aggregation of the endogenous alpha-syn both in SH-SY5Y cells but also in dopaminergic neurons.

This work should be discussed and potential differences that may lead to different findings should be suggested.

In its current form, I do not support publication of this paper in Communications Biology.

Raising this omitted citation is a good point; we have now referenced and discussed the work by Pinotsi et al in the revised manuscript. Our work detects seeded aggregation of endogenous α -syn without dye labeling. This therefore preserves the physiological structure and behavior of fibrils/monomers as closely as possible. Observation in α -syn aggregation in cells has been reported in a similar fashion in Pinotsi et al using external dye-labeled fibrils. With larger-sized seed fibrils, seed internalization carried out with 100 \times lower seed concentrations and without mediators, it is likely that the majority of the seeds stayed in the endosomal-lysosomal system as seen here, but lysosomal degradation was less likely to be overwhelmed by internalized seeds. In contrast, our current work sees the ALP overwhelmed, and this allows more seeds to escape to the cytosol, resulting in higher numbers of α -syn aggregates and a small apparent size of aggregates. The aggregates in the cytosol can be further fragmented by the proteasome and lead to even smaller aggregate sizes. In addition, the dye-labeled fibril seeds used by Pinotsi et al, contain a high proportion of unlabeled monomers. This can generate unlabeled seeds, especially small aggregates, leading to undetectable heterofibrils and hence a bias to the growth of larger aggregates. Our experiments can detect unlabelled endogenous protein, so we are able to measure time-dependent evolution of extracellular α -syn aggregates secreted by these cells, without aggregates being rendered undetectable by diluting out of extrinsically-labeled monomers.

We have added additional explanation of how these images are obtained in the main text and also explained why the aggregates are a collection of points by adding a new supplementary Figure 1. The aggregates that we observe are 30 nm in size making them hard to see in images of cells, but we have added more zooms of the aggregates formed to make their shape clearer as a supplementary figure.

In our experiments, only a handful of seeds enter each cell, so nearly all the aggregates formed after seeding are composed of endogenous α -syn.

In more detail, here are my comments and suggestions for improvement:

1. The papers reports on alpha-syn aggregation but there are no convincing nanoscopic images of aggregates, neither of the added fibrils nor of the induced aggregates. How do the authors distinguish between the added alpha-synuclein aggregates and the endogenous protein / induced aggregates?

The seeds are in very low quantity per cell as shown at T0 in Fig.1b and 1c. T0 (after seeding and rinsing with PBS) barely any aggregates are detectable and quantification shows nearly zero aggregates per cell. This clearly suggests that the seeds acquired by cells are very low in numbers; hence all the aggregates that form after seeding are due to aggregation of the monomer in the cell. We have added a sentence to explain this in the revised text.

2. According to the authors monomeric alpha-syn is not efficiently detected by antibody-DNA-PAINT imaging – does that hold for oligomeric too? That means that not all the effects on the aggregation and nucleation of the endogenous alpha-syn can be detected with this imaging method. The authors should also use a complementary super resolution imaging method with antibody labeling for example to image the endogenous protein, to support their findings.

In this work, we have used an aptamer and an antibody that specifically bind α -syn aggregates to specifically image α -syn aggregates by DNA paint, in a high excess of monomer. It is not possible to use an antibody to the monomer to also image the endogenous protein since it is present at too high concentrations in the cell, 10-20 μ M, to allow super-resolution. However, as discussed above, the aggregates formed after seeding are formed from the endogenous protein. We have added a sentence to clarify this in the paper.

3 . How is the counting of number given that what we see in the figure are "cluster-like"? What is the unit size of each counted species?

Each individual aggregates of which examples are shown in Figure 1 and the new supplementary figure are counted as individual aggregates and we report the number of these aggregates per cell.

4. Please define PFF.

We appreciate the reminder and will update the main text with a definition.

5. Figure 1: Poor quality and pixelated.

Panel b: the Bright Field Images have a strange bright "shading": die to improper illumination?

This panel is so small the reader cannot really "see" the aggregates and where they are located.

The problem is that the aggregates are very small compared to the cell, making them hard to see. We have provided more representative SR images of the cells in Figure 1 and S16,

to make the aggregate clearer as well as supplementary videos to show how the SR images are obtained. For Fig. 1a, as stated in the figure legend of Fig. 1a, the only pixelated greyscale image was recorded from thioflavin-T stained DL (diffraction-limited) images. The paired, colored image is super-resolved, which clearly demonstrates the high quality of images of our super-resolution techniques.

The shading the reviewer mentions might result from suspended cell debris. However, the brightfield images only provide cellular boundary information and do not alter any conclusions of the manuscript. We have replaced the images in Fig 1 with other representative images without shading.

6. Figure 2: in this image we see a lot of aligned, "linear" aggregates, compared to the previous one. What is the reason for this?

The aligned pattern (mostly along the plasma membranes) in general occurs in many cells of seeded samples, and the size/pattern varies between cells. These are many single aggregates that can be distinguished in the zoomed-in images. This phenomenon can be seen at later time stage from 24h to 72h when α -syn aggregates accumulate to higher levels. We did not carry out in-depth study in this regard, but this is in agreement with previous finding that in the presence of α -syn fibrillar seeds, the accumulation of exogenous α S fibrils takes place inside the cytoplasm within close proximity of the nucleus at early times (Luk, K.C. et al., PNAS, 2009). We have added a discussion of this point.

7. Figure 3: Panel b: Why at 24h there are more aggregates than 48h for the control?

There also seems to be no significant increase between 24h, 48h and 72h?

Why does the control also show increase of number of aggregates for 0, 4 and 24h?

Panel c: the control sample also has aggregates of the same size?

Panel d: it seems all aggregates in control and seeded have the same eccentricity. Why is that?

To 1st question: In Fig. 3b control of 24h and 48h do not have significant difference; SDs and 2-way ANOVA results are included to support this.

To 2nd question: it has been answered in #2 from reviewer 1. A steady state is reached where rate of aggregate production is balanced by aggregate removal.

To 3rd and 4th question: Our data in Figure 3 show that both the seeded and control cells reach a plateau in number of secreted aggregates and that these aggregates are similar in size and shape. The main difference is that the plateau is higher for the seeded cells. We can only speculate that this is because the control cells form and secrete aggregates to prevent any accumulation of aggregates in the cytosol over time. It takes time for this steady state to be established. If these cells are seeded, this steady state is disrupted, and the cells form and secrete more aggregates. But this secretion is limited and not sufficient to prevent the accumulation of aggregates in the cytosol with time. We have added a short paragraph discussing these points in the discussion, and thank the reviewer for raising this point.

8. Figure 4: Where are the super resolution images of the aggregates mentioned?

We have corrected the legend of Figure 4a with 'Number of aggregates per μm^2 after immunodepletion'.

9. In Materials and Methods: the pixel sizes of 96nm and 107 nm are large compared to the size of aggregates to be resolved.

Pixel size is not directly related to the resolution of super-resolved images. As described in the introduction and in the new supplementary Figure 1., by detecting single fluorophores, we know the fluorophore must be located at the centre of the fluorescent spot and can determine its location by fitting its point spread function. PAINT relies on fitting a point-spread function to the fluorescence from single fluorophores in each frame (raw image), then accumulating multiple frames to 'paint' out the shape of the object present. Because of diffraction, the emitted light of a single fluorophore is distributed over several hundred nanometers in each dimension and the pixels do not need to be of similar size to the object being super-resolved.

10. Why do the authors state that micromolar is the normal seeding concentration in cells? That is true for in vitro experiments.

We are unclear about which statement the reviewer refers to. If the reviewer refers to Discussion, second paragraph, last sentence, which is the closest match to the question, we would respond that this is sub-micromolar, not micromolar concentration that comes from the citation in the sentence, and this is clearly stated to be the regime in an iPSC-derived neuron. The current cell model of SH-SY5Y requires micromolar concentrations of α -syn seeds with the help of Bioporter. We cannot find any statement that micromolar is the normal seeding concentration in cells anywhere in the main text.

Reviewer #3 (Remarks to the Author):

Sang et al. report work on using super-resolution microscopy (SRM) to monitor α -syn aggregation in a model neuronal cell line. It is a follow up on the previous in vitro work from the same lab, and the results shed new light on the α -syn aggregation process, particularly the role of pre-existing α -syn aggregates in seeding new aggregates, the cellular machineries potentially involved in the processing and trafficking of α -syn aggregates. The manuscript is well organized, and the data are clearly presented and convincing. It is also a nice demonstration of how SRM could help resolve biological processes involving nanoscopic structures in intact cells, which would have been difficult with conventional fluorescence microscopy or with electron microscopy. I suggest that the authors consider the following when revising the manuscript.

a) The discussions section reads nicely. To help better keep track of what we have learned and what the remaining questions are, it'd be useful to add a figure that summarizes the results so far into a working model, hopefully also incorporating answers to (and/or hypotheses for) some of the questions below;

We appreciate the suggestion and have added a working model in a new Figure 5.

b) What types of α -syn structures (can) actually serve as seeds for growing new α -syn aggregates? The authors mentioned two different pre-existing α -syn fibrils used for seeding, sonicated fibrils or 'soluble' ones. They used sonicated fibrils and have done some characterizations on these in the present work. What do we know about the 'soluble' aggregates in terms of size, efficiency in cellular uptake and driving new α -syn formation once in cells? Also, can the small, soluble α -syn aggregates secreted by the cells be taken up by the cells in an autocrine manner and seed formation of α -syn aggregate formation? If so, this would yield a feed-forward process, in which case a braking mechanism would seem necessary. Correct me if I am mistaken here.

We did not carry out detailed seeding studies with soluble aggregates. Seeding competency of various sized α -syn aggregates has been previously studied in reference #49, which shows that competent α -syn seeds are $>75\text{nm}$. This suggests that the extracellular aggregates we saw - with an average size of $\sim 30\text{nm}$ - are unlikely to be effective seeds. This information is included in the discussion section of the revised manuscript.

c) The product of α -syn fibril sonication is likely a mixture of aggregates in different sizes, and it would be helpful to examine whether a subset of this mixture was preferentially taken up by the cells. I presume that the aggregates observed at $T=0\text{h}$ represent the original seeds, although some may have already been processed by the UPP and/or ALP (hence some of the data were already there)? Knowing this would be helpful to better define the seeding process.

The initial aggregates observed are very likely to be external seeds and partially processed by UPP/ALP. We agree that this is an interesting topic. Previous literature (cited as reference #42-44) are discussed. It has been shown that seeding results in accumulation of seeds in ALP hours after seed internalization. Since we also observed a deficit of ALP, it is highly likely that to be the same case. In this regard, the average length of $\sim 40\text{nm}$ at T_0 (Fig. 1d) and the population distribution (Fig. S6, largely distribute 30-100nm at T_0) indicates that the sonicated PFFs internalized in cells are shorter than original ones as shown in the TEM images (Fig. S1, sonicated, ~ 100 -150nm).

d) When imaging low abundance species such as the α -syn aggregates at $T=0\text{h}$ and $T=4\text{h}$, signals from the substrate (e.g. coverglass) could be problematic. The authors overcame this by switching from antibody-DNA-PAINT, which suffers more from nonspecific sticking to the coverglass but distinguishes α -syn aggregates better from $A\beta$ to using aptamers for DNA-PAINT (AD-PAINT), which recognize both α -syn aggregates and $A\beta$ but are less sticky to coverglass. Although not directly related to the biological questions, it would still be helpful for the authors to provide example raw images or videos in the supplementary material.

We have provided raw videos of Antibody-DNA-PAINT and AD-PAINT as Figure S16 and Video 1-4.

e) It would also be useful to include in the supplementary material schematics for imaging α -syn aggregates using either AD-PAINT or antibody-DNA-PAINT, for example how the actual dimensions of the aggregates are related to the apparent dimensions measured in the SRM images depending on the imaging strategy;

We have provided a new schematic for Antibody-DNA-PAINT and AD-PAINT as a new Fig. S1.

f) The legend for Figure 4 needs attention.

We have corrected the legend of Figure 4a with 'Number of aggregates per μm^2 after immunodepletion'.

REVIEWERS' COMMENTS:

Reviewer #1 (Remarks to the Author):

Im happy with the additional work and comments by the authors.

Reviewer #2 (Remarks to the Author):

I am happy to see that the authors have made substantial improvements to the manuscript, after the first revision round. However, there are important remaining issues that need attention and clarification. I believe acceptance of the paper should be considered only if the authors clearly address my following concerns/comments:

1. This is related to another reviewer's comment: Could you please show also the 'bioporter only' data? It is mentioned that these have been acquired, but not shown.
2. Related to the answer to my previous comment 2: Above which size of aggregates does the antibody bind to? It is still not clear to me whether or not the antibody binds to monomeric or oligomeric protein.
3. Related to the answer to my previous comment 7: What is the reason why the cells form spontaneously aggregates? It should be commented. Then if that is the case, why do the authors not detect any aggregate presence in the control examples in Figure S5 for PBS only and for the case of addition of monomeric protein?
4. Related to the answer to my previous comment 9: The new pixel size of the super resolved image is actually relevant, as when it is large then the fitted PSF of the emitter falls within a much larger area. I would recommend to reduce the pixel sizes of the super resolved images to a size of equal or smaller than the mean localization precision.
5. Related to the answer to my previous comment 10: The authors state this in page 8 of the new Suppl Information.
6. Could you show a super-resolved image (DNA -PAINT) of the in vitro sonicated fibrils (PFF) before addition to cells to compare the sizes with the ones after addition at $T=0$? At the moment only TEM images are shown. It would help if the authors would add the length distribution for the PFFs in figure S9.
7. In Figure 1a: if this is one aggregate why is it shown as '2 dots', instead of a continuous structure? Would that be counted as one or two?
8. Supplementary videos: What are the bright spots in the movies? Their width is 8 pixels corresponding to 0.8 μ m if I interpreted the scale bar correctly. However the fluorescent beads should be only 100nm.

Reviewer #3 (Remarks to the Author):

The authors have addressed my comments adequately. Their efforts to address other reviewers have resulted in additional figures, suppl data, and discussions, which I find useful. I therefore recommend that the paper be accepted for publication.

REVIEWERS' COMMENTS:

Reviewer #1 (Remarks to the Author):

Im happy with the additional work and comments by the authors.

We thank Reviewer 1 for the support of our work.

Reviewer #2 (Remarks to the Author):

I am happy to see that the authors have made substantial improvements to the manuscript, after the first revision round. However, there are important remaining issues that need attention and clarification. I believe acceptance of the paper should be considered only if the authors clearly address my following concerns/comments:

1. This is related to another reviewer's comment: Could you please show also the 'bioporter only' data? It is mentioned that these have been acquired, but not shown.

We have now included this negative control in supplementary materials, and we have updated SI with this data (Fig. S7b).

2. Related to the answer to my previous comment 2: Above which size of aggregates does the antibody bind to? It is still not clear to me whether or not the antibody binds to monomeric or oligomeric protein.

The conformational antibody (MJFR-14-6-4-2) preferentially recognizes aggregated α -syn, both fibrils and oligomers. This is shown in the manufacturer's description and we directly tested this with α -syn monomer-seeded cells. There has also been a detailed study of binding of this antibody and several other ones to monomer, oligomers and fibrils [Kumar et al, *Neurobiology of Disease* **146**, page 105086 (2020)]. This study also confirmed that the MJFR-14 antibody bound oligomers, 10 nm in size, and fibrils but not monomers. Thus, the MJR-14 antibody binds aggregates 10 nm or larger.

The above citation has been added to References.

3. Related to the answer to my previous comment 7: What is the reason why the cells form spontaneously aggregates? It should be commented.

a-syn is present in the cell at a concentration of about 20 μ M. Some of this a-syn may be bound by chaperone or lipid membranes and not available to aggregate reducing the effective concentration. However, we have previously shown that a-syn at a concentration as low as 0.5 μ M can spontaneously aggregate into small oligomers [Iljina et al, *PNAS* **113**(9), page E1206-E1215 (2016)] so we presume that this process is also taking place in the cell.

We have now briefly commented on this in the paper.

Then if that is the case, why do the authors not detect any aggregate presence in the control examples in Figure S5 for PBS only and for the case of addition of monomeric protein?

The aggregates produced in the control experiments are secreted by the cells as fast as they form, so that no aggregates are detected in the cells but aggregates can be detected in the culture media.

4. Related to the answer to my previous comment 9: The new pixel size of the super resolved image is actually relevant, as when it is large then the fitted PSF of the emitter falls within a much larger area. I would recommend to reduce the pixel sizes of the super resolved images to a size of equal or smaller than the mean localization precision.

The super-resolved images are already presented with a pixel size of 13 nm, smaller than the mean localisation precision in our paper.

Since each molecule is represented as a Gaussian with a standard deviation equal to the localisation precision, the pixel size used needs to be smaller than the mean localisation precision to define the Gaussian shape and this is standard in the field.

5. Related to the answer to my previous comment 10: The authors state this in page 8 of the new Suppl Information.

Our original text is: 'Since there are 5×10^4 cells per cm^2 during seeding with an area of wells of 9.6 cm^2 , this is roughly $2 \times 10^7 - 2 \times 10^9$ seeds per cell, hence a large excess of seeds over cells. This is the normal situation in most cell seeding experiments.'

We have modified this to clarify that we meant a large excess of seeds over cells is the normal situation in most cell seeding experiments, not micromolar concentrations. The sentence now reads: 'Since there are 5×10^4 cells per cm^2 during seeding with an area of wells of 9.6 cm^2 , this is roughly $2 \times 10^7 - 2 \times 10^9$ seeds per cell, hence a large excess of seeds over cells. This large excess of seeds over cells is the normal situation in most cell seeding experiments.'

6. Could you show a super-resolved image (DNA -PAINT) of the in vitro sonicated fibrils (PFF) before addition to cells to compare the sizes with the ones after addition at $T=0$? At the moment only TEM images are shown. It would help if the authors would add the length distribution for the PFFs in figure S9.

We believe the right panel of Fig. S3a already shows PFF image using AD-PAINT; left bar chart of Fig. S3b is the length distribution of PFFs.

7. In Figure 1a: if this is one aggregate why is it shown as '2 dots', instead of a continuous structure? Would that be counted as one or two?

Figure 1a aims to show the advantage of super-resolution imaging. It shows the shapes of the two aggregates can only be fully resolved when they are imaged at super-resolution. In contrast, in a diffraction-limited image you cannot resolve them and will end up with only one blurred spot.

We have now modified Figure 1a and the legend to make this clear.

8. Supplementary videos: What are the bright spots in the movies? Their width is 8 pixels corresponding to 0.8 μ m if I interpreted the scale bar correctly. However the fluorescent beads should be only 100nm.

The bright spots are fiducial markers (TetraSpeck 0.1- μ m fluorescent microspheres) as described in the Methods. These also appear as diffraction limited spots with a Gaussian profile and since they are very bright, fluorescence signal is detected over many more pixels than the dim molecules. This allows us to fit the PSF much better and more precisely determine the location of the centre of the bead to correct for drift.

We have added that the bright spots are fiducial markers in the description of the videos .

Reviewer #3 (Remarks to the Author):

The authors have addressed my comments adequately. Their efforts to address other reviewers have resulted in additional figures, suppl data, and discussions, which I find useful. I therefore recommend that the paper be accepted for publication.

We thank Reviewer 3 for the support of our work.